# *Ripply3* overdosage induces mid-face shortening through *Tbx1* downregulation in Down syndrome models

**José Tomás Ahumada Saavedra**[1], **Claire Chevalier**[1], **Agnes Bloch Zupan**[1,2,3,4],
**Yann Herault**[1]*

**1** Université de Strasbourg, Institut de Génétique et de Biologie Moléculaire et Cellulaire (IGBMC), INSERM U1258, CNRS- UMR7104, Illkirch, France, **2** Hôpitaux Universitaires de Strasbourg (HUS), Pôle de Médecine et Chirurgie Bucco-dentaires, Centre de Référence des maladies rares orales et dentaires, CRMR-O-Rares, Filière Santé Maladies rares TETE COU and European Reference Network ERN CRANIO, Strasbourg, France, **3** Université de Strasbourg, Faculté de Chirurgie Dentaire, Strasbourg, France, **4** Université de Strasbourg, Institut d'études avancées (USIAS), Strasbourg, France

* herault@igbmc.fr

## Abstract

The most frequent and unique features of Down syndrome (DS) are learning disability and craniofacial (CF) dysmorphism. The DS-specific CF features are an overall reduction in head dimensions (microcephaly), relatively wide and broad neurocranium (brachycephaly), reduced mediolaterally orbital region, reduced bizygomatic breadth, small maxilla, small mandible, and increased individual variability. Until now, the cellular and molecular mechanisms underlying the specific craniofacial phenotype have remained poorly understood. Investigating a new panel of DS mouse models with different segmental duplications on mouse chromosome 16 in the region homologous to human chromosome 21, we identified new regions and the role of two candidate genes for DS-specific CF phenotypes. First, we confirmed the role of *Dyrk1a* in the neurocranium brachycephaly. Then, we identified the role of the transcription factor *Ripply3* overdosage in midface shortening through the downregulation of *Tbx1*, another transcription factor involved in the CF midface phenotype encountered in DiGeorge syndrome. This last effect occurs during the development of branchial arches through a reduction in cell proliferation. Our findings define a new dosage-sensitive gene responsible for the DS craniofacial features and propose new models for rescuing all aspects of DS CF phenotypes. This data may also provide insights into specific brain, immune and cardiovascular phenotypes observed in DiGeorge and DS models, opening avenues for potential targeted treatment to soften craniofacial dysmorphism in Down syndrome.

**Data availability statement:** The morphological data are deposited in Zenodo at https://doi.org/10.5281/zenodo.13639386

**Funding:** This work of the Interdisciplinary Thematic Institute IMCBio, as part of the ITI 2021-2028 program of the University of Strasbourg, CNRS and Inserm, was supported by IdEx Unistra (ANR-10-IDEX-0002), SFRI-STRAT'US project (ANR 20-SFRI-0012), INBS PHENOMIN (ANR-10-INBS-07) and EUR IMCBio (ANR-17-EURE-0023) under the framework of the French Investments for the Future Program. YH received funding from the European Union's Horizon 2020 research and innovation program under grant agreement No 848077 for GO-DS21. JTAS received funding from the National Agency for Research and Development (ANID)/Scholarship Program/DOCTORADO BECAS CHILE/2020- 72210028. The funders had no role in study design, data collection and analysis, decision to publish, or preparation of the manuscript.

**Competing interests:** The authors have declared no competing interests exist.

## Author summary

An extra copy of chromosome 21 causes Down syndrome (DS) and leads to intellectual disability and distinct facial features. To understand which genes drive these traits, we have engineered mouse models that mimic the DS genetic changes. By studying these mice, we identified regions of mouse chromosome 16, equivalent to human chromosome 21, that are associated with the head and face differences in Down syndrome (DS). We pinpointed two genes, *Dyrk1a* and *Ripply3*, whose increased activity disrupts normal facial development by altering the growth of specific cells during embryonic development. Notably, reducing Ripply3 activity in these mice corrected some facial abnormalities, highlighting its crucial role. These findings clarify how extra genetic material leads to DS features and suggest new therapeutic targets. The study also highlights the importance of animal models in elucidating the genetic and developmental basis of human conditions, such as DS. Additionally, the mechanism involving *Ripply3* connects DS with DiGeorge syndrome, another disorder with similar facial changes, suggesting shared pathways behind some features of both conditions.

## Introduction

Trisomy 21, or Down syndrome (DS), is a pleiotropic disorder with intellectual disability, CF changes, and comorbidities. Somehow, gene dosage effects of one or more of the 222 genes on human chromosome 21 (Hsa21) are responsible for specific pathological features [1,2]. Facial features are characteristic of individuals with DS [3], although their severity varies from one individual to another [4]. DS-related CF dysmorphism includes an overall reduction in head dimensions (microcephaly), relatively short and broad neurocranium (brachycephaly), small midface, reduced mediolaterally orbital region, reduced bizygomatic breadth, small maxilla, and small mandible [5,6,7,8]. Patients also experience a low bone mass associated with reduced osteoblast activity and high bone turnover [9].

Studies in rodents and humans have attempted to identify the candidate gene(s) causing DS clinical features [10,11]. Using the rapid engineering of the *Mus musculus* (Mmu) genome, multiple DS mouse models have been generated [12,13] containing extra copies of the Hsa21-orthologous regions of three murine chromosomes: Mmu chromosome 16 (Mmu16), 10 (Mmu10), and 17 (Mmu17) [14,15].

DS mouse models have previously been studied for CF phenotypes. The most notable studies have examined the Ts(17[16])65Dn model, hereafter termed Ts65Dn. This mouse strain carries an extra mini-chromosome with the *mIR155-Zbtb21* region of Mmu16 translocated downstream of *Pde10a,* close to the centromere of Mmu17 [16]. Thus, Ts65Dn is trisomic for 104 of the Hsa21 orthologs of the Mmu16 between *miR155 and Zbtb21* [17,18,19]. The Ts65Dn model displays a variety of phenotypes similar to those found in DS individuals [17], including a low bone mass caused by intrinsic cellular defects in osteoblast differentiation, reducing bone formation [20]. In

addition, bone resorption mediated by osteoclasts is also reduced, but this is not enough to overcome the low rate of bone formation [21,22]. These animals also show many cognitive and behavioral traits as well as characteristic skeletal, craniofacial, cardiovascular features, granule-cell density of the dentate gyrus, and megakaryocytopoiesis mimicking the phenotype encountered in individuals with Down syndrome [8]. The CF phenotypes found include brachycephaly, reduced facial and cranial vault dimension, reduced cerebellar volume, and many features present in individuals with DS. CF changes even more similar to those in humans were found in our new Ts(17[16])66Yah model, devoid of non-Hsa21 triplicated genes [23], confirming a significant contribution of one or more genes found between *mIR155* and *Zbtb21* to the CF phenotypes.

CF defects have also been detected in other DS mouse models. The Ts(16C-tel)1Cje (Ts1Cje) model carries a translocation that encompasses 81 orthologous genes between *Sod1* and *Mx1* [15,24] and displays a generalized reduction in CF size with additional features [25]. By contrast, the Dp(16*Cbr1-Fam3b*)1Rhr (noted here Dp1Rhr), a model trisomic for 33 genes [26], exhibited a larger overall size and CF alterations, including more pronounced defects in the mandible than observed in Ts65Dn mice and individuals with DS [27].

The Dp(16*Lipi-Zbtb21*)1Yey mouse model (Dp1Yey) is a larger model with a 22.9 Mb direct duplication of the entire Mmu16 region in conserved synteny with Hsa21, containing 118 orthologous protein-coding genes [28]. The CF phenotype corresponds to brachycephaly, a reduced dimension of the maxillary and palate, and reduced mandibular size. The skulls also exhibited increased variance relative to euploid littermates for specific linear distances [28,29,30].

Another model quite similar to Dp(16)1Yey, but with a slightly different duplicated interval, is the Dp(16*Lipi-Zbtb21*)1 TybEmcf, or Dp(16)1Tyb [31]. In 2023, using morphometric analysis of the Dp1Tyb mouse model of DS and an associated mouse genetic mapping panel [ 32], showed that *Dyrk1a* is required in three copies to cause CF dysmorphology in Dp(16)1Tyb mice. In addition, Dp(16)1Tyb mice display many phenotypic features characteristic of DS in humans, including congenital heart defects, reduced bone density, and deficits in memory, locomotion, hearing, and sleep [33,34,35,36]. Taken together, the candidate genes responsible for craniofacial phenotypes found in DS models include *Dyrk1a*, *Rcan1* (*Dscr1*), and *Ets2*. *Dyrk1a* has been implicated in several DS phenotypes, including cognitive impairment, motor function, and craniofacial abnormalities [37,38,39]. Johnson et al. in 2024 [40], showed that a decreased in *Dyrk1a* in Xenopus resulted in craniofacial malformations, altered expression of critical craniofacial regulators as *Pax3* and *Sox9* fundamental for cranial neural crest development, and presented altered retinoic acid, hedgehog, nuclear factor of activated T cells (*NFAT*), *Notch* and *WNT* signaling pathways. These results indicate that DYRK1A function is critical for early craniofacial development and must properly regulate the expression of specific craniofacial regulators in the branchial arches [40].

Disruption of *Tbx1* expression is a common aspect of CF dysmorphias. *Tbx1* Is the first dosage-sensitive gene identified in the DiGeorge syndrome (DGS)/velocardiofacial syndrome (VCFS), a congenital disorder characterized by neural-crest-related developmental defects. In humans, *TBX1* haploinsufficiency causes craniofacial anomalies [41]. In the mouse model for DiGeorge syndrome the CF phenotype observed in the mutant mice for the T-box gene, *Tbx1*[+/-], encompasses abnormal development of the skeletal structures derived from the first and second pharyngeal arches, with reduced dimension of the midface [42]; a similar situation found in the DS mouse models.

However, the details of how the dosage imbalance of Hsa21 genes affects CF morphogenesis are still poorly understood [43]. Unravelling the complex genetics and adaptive biological processes involved in forming craniofacial structures is essential. Many genes are conserved across mammals, implying that the genetic programs for a specific phenotype may also be conserved. Therefore, this can validate the study of animal models to decipher human genetic outcomes [44].

As observed in various models, human partial trisomy has enabled the mapping of genetic regions on Hsa21 that contribute to CF anomalies. Still, a specific region has not yet been identified [45,10]. Identifying the dosage-sensitive genes responsible for each element of the DS phenotype will help us better understand the molecular mechanisms underlying the various symptoms and will enable us to define more effective therapeutic options [46,47].

The Cre-LoxP technology has enabled the engineering of more precise duplications [48,49]. Applying this technology, we generated mouse models carrying different segmental duplications of regions located on the Mmu16 homologous to

Hsa21. In this study, we utilised these new DS models and two previously known models, Dp(16)1Yey and Tg(Dyrk1a), to establish correlations between human-related CF phenotype and genotype and to understand the potential craniofacial effect of the duplication of different chromosomal regions via a morphometric analysis of the animal models. This led us to narrow our research to identify new Mmu16 regions involved in CF and find corresponding candidate genes responsible for the DS-CF phenotype.

## Materials and methods

### Ethics statement

All the experiments were done in our facility approved for breeding and using animals for scientific purposes under number: D 67-218-40 (France) and under supervision by the Ethical committee COM'ETH n°17, following the European Directive (2010 /63/EU) and the Decree No. 2013–118 of February 1, 2013: concerning the protection of animals used for scientific purposes in France.

### Previously reported rodent models used

The Dp(16)1Yey and Tg(Dyrk1a) (official name Tg(Dyrk1a)189N3Yah) models [28,50] were maintained on the C57BL/6J genetic background. We also used the SD:CRL Dp (11Lipi-Zbtb21)1Yah (short name Dp(Rno11)) rat model generated in the lab [51] that carries a duplication of the Lipi-Zbtb21, an interval similar to the mouse Dp(16)1Yey, found on rat chromosome 11.

### Generation of the new DS mouse strains

These new lines were generated via an *in vivo* chromosomal recombination technique, which combines a transposon system [49] and a meiotic recombination system Cre-LoxP [52]. The transposon system consists of the transposase enzyme and its substrate, the transposon. The enzyme recognizes specific repeat sequences (ITR) flanked on either side of a given DNA sequence (in this case, a vector containing a specific *loxP* site) [49]. Briefly, a region of interest with two *loxP* sites inserted, one on each homologous chromosome, and a transgene expressing the Cre recombinase is brought into the same individual by successive crosses. In this animal, the Cre enzyme recombines the sequences of the *loxP* sites to produce a duplication (or partial trisomy) of the region of interest [52,48].

The new mouse models have been developed with the following segmental duplications in the Mmu16. For Dp(16Samsn1-Cldn17)7Yah (Dp(16)7Yah) we duplicated the segment between *Samsn1* and *Cldn17*. Dp(16 Tiam1-Clic6))8Yah (Dp(16)8Yah) presents a duplication between *Tiam1* and *Clic6*. Dp(16Cldn17-Brwd1))14Yah (Dp(16)14Yah) displays a duplication in the interval between *Cldn17* and *Brwd1*. Dp(16Tmprss15-Setd4)10Yah (Dp(16)10Yah) has the segment between *Tmprss15* and *Setd4* duplicated, similar to Dp(16Tmprss15-Grik1)11Yah (Dp(16)11Yah), but this model presents a region duplicated until *Grik1*. Dp(16Tmprss15-Zbtb21)12Yah (Dp(16)12Yah) has the duplicated region from *Tmprss15* to *Zfp295*, and Dp(16Cldn17-Vps26c(Dyrk1aKO))13Yah (Dp(16)13Yah) from *Cldn17* to *Vps26c*, up to the sequence of *Dyrk1a* which is inactivated. All lines were maintained on C57BL/6J genetic background (Fig 2A).

### Adult cohorts generated

Mice were housed under specific pathogen-free (SPF) conditions and were treated in compliance with the animal welfare policies of the French Ministry of Agriculture (law 87 848). For each mouse line, about ten littermates of each genotype, DS, and wild-type (WT) were collected after euthanasia using a standard, validated and approved humane killing procedure (n = 180 in total) by competent, adequately educated and trained staff. We attempted to balance males and females in the cohorts. As a major genotype effect compared to sex was previously described elsewhere independently [32], and

optimize the number of animals used in this project, to follow the 3Rs rule, we decided to check whether the sex effect was insignificant compared to genotype in the Dp(16)1Yey line. Thus, For the Dp(16)1Yey line, six females plus five males for the dup carrier and six males plus three females for control were used. As we did not find sexe as a significant variable, we then continued mainly without considering sexe, with sometimes more female individuals collected than males, because males were used to breed the lines.

## Micro-computed tomography scan of the skull of mutant and control mouse lines

Animals were euthanized with the standard procedure at 14 weeks old. Briefly, the mouse heads were dissected apart from the body. A polystyrene section was interposed between the mandible and maxilla to separate the jaws. After dissection, samples were fixed in a 4% paraformaldehyde solution (PFA), washed with water, and stored in 70% ethanol. The mouse heads were scanned using the Quantum FX micro-computed tomography imaging system (Caliper Life Sciences, Hopkinton, MA, USA) to evaluate the morphology of the skull and mandible. The images obtained were delivered in DICOM format. The scan parameters used to carry out the scanning of the samples correspond to 2 scans of every sample, anterior part, and posterior part using the mode Scan Technique Fine of 2 minutes, with a field of view (FOV) of 40 mm, the voltage 90 kV, CT 160 μA, resolution pixel size 10 μm and the capture size for live mode viewing in small, live current 80kV.

## Imaging processing

For each sample, two scans were obtained, one from the anterior area of the skull and one from the posterior region. FIJI software was used to unite these two scans and create a single file, performing the plugin "Stitching" and making one file in TIFF format. This format can be opened using different image processors. Stratovan Checkpoint software (Stratovan Corporation, Sacramento, USA, Version 2018.08.07. Aug 07, 2018.) was used to place the landmarks (S1 and S2 Tables), extract the 3D coordinates, create Procrustes average models, and perform the voxel analysis. 3dMD Vultus software (3dMD LLC, Atlanta, GA, USA) generated heat maps.

## Morphometrics analysis

Morphometrics is the quantification and statistical analysis of form. Form is the combination of size and shape of a geometric object in an arbitrary orientation and location (shape is what remains of the geometry of such an object once it is standardized for size). Various approaches can be employed when conducting morphometric analysis. The method of interest in this study is the landmark-based method, which is a conventional approach that relies on phenotypic measurements such as linear distances, angles, weights, and areas. In this case, we used 61 landmarks, 39 in the skull and 22 in the mandible, to obtain the 3D coordinates of the structure [53].

Based on 3D coordinates, Euclidean Distance Matrix Analysis (EDMA) is one of the principal tools for analyzing landmark-based morphometric data [54]. This method builds a matrix of linear distances between all possible pairs of landmarks for each specimen [55]. Morphological differences between groups can be pinpointed to specific linear distances on an object through pairwise comparisons of mean form or shape matrices, followed by bootstrapping to estimate the significance of these differences [54]. In this study, two tests were done for each group of samples, first to analyze the form of the skull and mandibles with form difference matrix (FDM) and then the shape with the shape difference matrix (SDM).

In addition, to track the landmarks associated with a significant change and understand where they are located in the CF structures, "EDMA FORM or SHAPE Influence landmark analysis" was performed [56]. The purpose of this test is to search which landmarks present a Relative Euclidean distance (RED) > 1.05 or < 0.95 (outside of the confidence interval 97,8%), meaning, which landmarks show a bigger difference in linear distances between every landmark and in what direction.

Another way to handle landmark-based data is using a multivariate statistical analysis of form, geometric morphometric. This method relies on the superimposition of landmark coordinate data to place individuals into a common morpho-space. The most used superimposition form is the Generalized Procrustes (GP) method and Principal Component Analysis (PCA). This method places multiple individual specimens into the same shape space by scaling, translating, and rotating the landmark coordinates around the centroid of every sample [57]. As an alternative, we took advantage of Stratovan Checkpoint (Stratovan Corporation, Sacramento, USA) to create population average models and perform a voxel-based analysis, where we can observe directly in 3D models the changes between populations. The analysis used the average WT model as a reference; therefore, the changes were observed in the average DS model analyzed. Indicated in red are the structures with a higher dimension in the DS model of interest, and in blue the structures with a lower dimension.

Finally, using the 3dMD Vultus software, we created Procrustes average models created in Checkpoint to perform a landmarking calculation.

## Whole-mount skeletal staining

DS individuals also experience a low bone mass associated with reduced osteoblast activity and bone turnover [9]. To study these defects in ossification, we performed a skeletal/cartilage staining with alizarin red and alcian blue in a representative DS mouse model, Dp(16)1Yey.

Whole-mount skeletal staining permits the evaluation of the shapes and sizes of skeletal elements. Thus, it is the principal method for detecting changes in skeletal patterning and ossification. Because cartilage and bone can be distinguished by differential staining, this technique is also a powerful means to assess the pace of skeletal maturation.

We collected n = 20 specimens, ten samples (5 WT vs 5 Dp(16)1yey) for the embryonic stage (E) 18.5 and 10 samples (5 WT vs 5 Dp(16)1yey) for P2 (2 days after birth), and were prepared by removing skin, organs, and brown fat. Then, they were dehydrated and fixed in 95% ethanol for four days. To further remove fatty tissue and tissue permeabilization, specimens were exposed to acetone for one day. Consecutively, samples were transferred to Alcian blue and Alizarin red staining solutions. Later, they were exposed to potassium hydroxide (KOH) for three days, leading to tissue transparency. Finally, they were preserved in Glycerol 87%. The procedure can be adjusted depending on the size/age of the specimens [58].

## Dissection of branchial arches during development, RNA extraction, and RT-digital droplet PCR (RT-ddPCR)

To study the level of expression of different genes triplicated on the DS mouse models, we performed RT-ddPCR (BioRad, Hercules, USA), a digital PCR used for absolute quantification that allows the partitioning of the cDNA samples obtained from the RT procedure up to 20,000 droplets of water-oil emulsions in which the amplification was performed [59,60].

For this, we collected the embryos of four pregnant females, Dp(16)1yey at E11.5 and four pregnant females of Dp(RNO11) (DS rat model with a complete duplication of chromosome 11) at E12.5. We obtained 20 embryos for each line (10 WT vs. 10 DS model, n = 40) and dissected the frontonasal process, maxillary process, mandibular process, lateral and medial nasal process, and first pharyngeal arch. The dissected tissues are placed in cryogenic storage vials and quickly transferred to liquid nitrogen to avoid RNA decomposition. RNA extraction was performed using the RNeasy Plus Mini Kit (QIAGEN, Hilden, Germany), and RNA quality and concentration were assessed using Nanodrop (Thermo Fisher Scientific, Illkirh, France). cDNA synthesis was performed using the SuperScript VILO cDNA synthesis kit (Invitrogen), the final reaction is diluted five times and stored at −20 °C until use.

For the PCR reaction, 2 µL of the diluted cDNA samples are supplemented with 10 µL of Supermix 2X ddPCR (without dUTP, Bio-Rad, #1863024), 1 µl of target probe (ZEN FAM)/primers mix (final concentration of 750 nM of each primer and 250 nM of probe) and 1 µl of reference probe (ZEN HEX)/primer mix (final concentration of 750 nM of each primer and 250 nM of probe) obtaining a total volume of 20 µl. Once prepared, the samples are fractionated into droplets using the QX200 droplet generator (Bio-Rad). The PCR reaction can then be performed by transferring 40 µL of the samples to a

96-well plate. The fluorescence intensity of each droplet is then measured with the QX200 reader (Bio-Rad). Data are analyzed using Quantasoft Analysis Pro software (Version 1.0.596). More detailed protocol can be found in [59].

Primers marked with specific fluorescent probes were designed using the PrimerQuest online web interface from IDT (https://www.idtdna.com/Primerquest/Home/Index) to target the genes of interest plus the housekeeping gene. Primers are blasted on the target gene map to verify that they span the exon/exon boundaries on the RNA. For mice and rats, the target genes were *Ripply3, Tbx1*, and *Dyrk1a*. The housekeeping gene for mice was *Tbp,* and for the rats, *Hprt1*. The primers are described below.

The primers used for mice were for *Ripply3* (ENSMUSG00000022941.9),forward AACGTCCGTGTGAGTCTTG, Reverse CTTTACTTACCCGTTTCAAAGCG, Probe ACACACATCGGGATCAAAGGGAGC (HEX); *Tbx1* (ENS-MUSG00000009097.11), forward CTGTGGGACGAGTTCAATCA, reverse ACTACATGCTGCTCATGGAC, probe TCAC-CAAGGCAGGCAGACGAAT (FAM); *Dyrk1a* (ENSMUSG00000022897.16), forward GCAACTGCTCCTCTGAGAAA, reverse AACCTTCCGCTCCTTCTTATG, probe AAGAAGCGAAGACACCAACAGGGC (HEX); and housekeeping gene *Tbp* (ENSG00000112592.15), forward AAGAAAGGGAGAATCATGGACC, reverse GAGTAAGTCCTGTGCCGTAAG, probe CCTGAGCATAAGGTGGAAGGCT (FAM/HEX).

For the rat samples, primers were for *Ripply3* (ENSRNOG00000001684.7),forward GCTGATCTGACCAGAACT-GAA, Reverse CGCTTTGAAATGGGCAAGTAA, Probe TTGGGAGGACCAACAAACCTTGGG (HEX); *Tbx1* (ENSRNOG00000001892.6), forward CAGTGGATGAAGCAGATCGTAT, reverse GGTATCTGTGCATGGAGTTAAGA, probe TCGTCCAGCAGGTTATTGGTGAGC (FAM); *Dyrk1a* (ENSRNOG00000001662.8), forward ACAGTTCCCATCA TCACCAC, reverse TCCTGGGTAGAGGAGCTATTT, probe AATTGTAGACCCTTGGCCTGGTCC (HEX); and house-keeping gene *Hprt1* (ENSRNOG00000031367), forward TTTCCTTGGTCAAGCAGTACA, reverse TGGCCTGTATCCAA-CACTTC, probe ACCAGCAAGCTTGCAACCTTAACC (FAM/HEX).

## EdU labeling

Depending on their capacity to proliferate, cells in the organism can be divided into three categories: proliferating cells, non-proliferating cells that have left the cell cycle, and quiescent cells capable of entering the cell cycle if necessary [61]. Proliferating cells continue to progress through the different phases of the cell cycle (G1->S->G2->M). Daughter cells from a previous division immediately enter the next cell cycle. There are several specific markers for each phase of the cell cycle. For example, Phosphohistone 3 (PH3) corresponds to phosphorylated histone 3 and is found in mitotic cells. EdU is a thymidine analog that can be incorporated into DNA during replication (S phase of the cell cycle). We can define the percentage of proliferating cells and cell cycle progression using these two markers.

First, pregnant females Dp(16)1Yey at E8.5 stage were injected intraperitoneally with EdU (SIGMA, ref. E9386; diluted in NaCl 0.9%, final concentration 7.5μg/μL; volume injected: 41ug for each milligram of animal weight). Then, embryos were collected 24 hours after EdU injection (E9.5). After, embryos were fixed with 4% paraformaldehyde and embedded in Shandon Cryomatrix Frozen Embedding Medium (Thermo Scientific). Frozen sagittal sections (14 μm) were cut using a Leica CM3050 S cryostat and placed on Superfrost Pluse slides for immunohistochemistry.

The immunohistochemistry for EdU was performed following the protocol described in the Kit de cellular proliferation EdU Click-iT for imaging, Alexa Fluor 555 (REF: C10338). For PH3 we used as primary antibody the Anti-phospho-Histone H3 (Ser10) Antibody, Mitosis Marker (Merck Millipore, REF: 06–570, 1:500) and secondary antibody the Don-key anti-Rabbit IgG (H + L), Alexa Fluor 647 (Invitrogen, REF: A-31573, 1:500). For the nuclear marker we used Hoechst nucleic acid stain (Invitrogen, REF: H3570, 1:2000). Quantitative analyses comparing wild-type and mutant embryos. The percentage of EdU-positive cells determined the proliferation index (the number of proliferating cells relative to the total number of cells, labeled with Hoechst) in the area of interest. In addition, the ratio of EdU-positive/PH3-positive cells allow to assess how many proliferating cells have progressed from S phase to G2/M phase (mitotic cells), thus defining the mitotic index.

## Results

### Contribution of *Lipi-Zbtb21* region to DS craniofacial features in Dp(16)1Yey mouse model

In individuals with DS, sexual dimorphism is observed from an early age, with higher measurements in males than in females, but the growth rate remains unchanged. However, in the studies where cephalometric superimposition variables were analyzed, these differences did not appear. This may be due to the low magnitude of the superimposition measurements, making it challenging to determine significant differences [62,63].

Previous studies in mice have shown no sex differences in the shape of the cranium and only a subtle difference in the shape of the mandible [64]. Importantly, for both the cranium and mandible, the effect of genotype was more substantial than sex for Dp1Tyb mice and the other strains [32]. Considering this information, we decided to verify this observation and used both sexes together in the CF analysis of Dp(16)1Yey.

First we analyzed the Dp(16)1Yey DS mouse models to compare them with the other DS model Dp(16)1Tyb [32]. Then, we performed morphometric analysis of Dp(16)1Yey on adult samples. We observed significant changes in form and shape difference matrix that can be understood as an overall reduction of dimensions (microcephaly) and smaller mandible (S1 Fig). For a more detailed investigation of the patterns of landmark displacements and their dimensionality, we employed principal component analysis (PCA). In the skull and mandible, the PCA of Dp(16)1Yey showed significant differences in the dimensionality versus the WT group (Fig 1A).

To track the landmarks associated with a significant change and understand where they are located in the CF structures, "EDMA FORM or SHAPE Influence landmark analysis" was performed [56]. The purpose of this test is to search for which landmarks present a Relative Euclidean distance (RED) > 1.05 or < 0.95 (outside of the confidence interval 97,8%), meaning which landmarks show a bigger difference in linear distances between every landmark and in what direction. On one side, the most influential landmarks with decreased dimensions corresponded to the ones from the maxillary bones, mandible, premaxilla, frontal, temporal (with the squamosal portion), and occipital bone. On the other side, landmarks with a significant increase in dimensions were in the cranial vault, the parietal bone, and the intraparietal bone (S1 Fig).

Additionally, by using Procrustes average models of the different populations to perform voxel analysis, we found that the key aspects of the Dp(16)1Yey phenotype correspond to a decrease in the dimensions of the midface, indicating midface hypoplasia and a short nasal region. In the neurocranium, an increase in dimensions in the lateral width was found, with a reduction in the occipital bone, leading to a shortening of the anteroposterior axis (brachycephaly). In the case of the mandible, we found a decrease in the width of the ramus, body, incisor alveolus, and molar alveolus and increased lateral dimension in the coronoid and condylar process (expected by the skull brachycephaly; Fig 1A).

Additionally, individuals with DS also experience low bone mass, associated with reduced osteoblast activity and decreased bone turnover [9]. Knowing this, along with the information obtained from the craniofacial analysis, we proposed that these significant changes could affect bone ossification during development. To address this, we performed a skeletal/cartilage staining with alizarin red and alcian blue in Dp(16)1Yey in embryonic stage (E) 18.5 and P2. At E18.5, mutant embryos exhibit a defect in mineralisation in the parietal bones, intraparietal, nasal bone, and atlas compared to WT (Fig 1B and 1C). Interestingly, no more phenotype was observed at P2 (Fig 1D). Altogether, the origin of the CF morphological changes in the Dp(16)1Yey are similar to Dp(16)1Tyb [64] and probably originates during pre-natal development in the mouse.

### Dissection of CF phenotype: Mapping the location of dosage-sensitive genes inside *Lipi-Zbtb21,* that cause the craniofacial dysmorphology using a new panel of mouse models

To elucidate the location of dosage-sensitive genes that are predominantly involved in the craniofacial dysmorphology of Dp(16)1Yey mice, we took advantage of a new panel of 7 mouse models with shorter segmental duplications covering

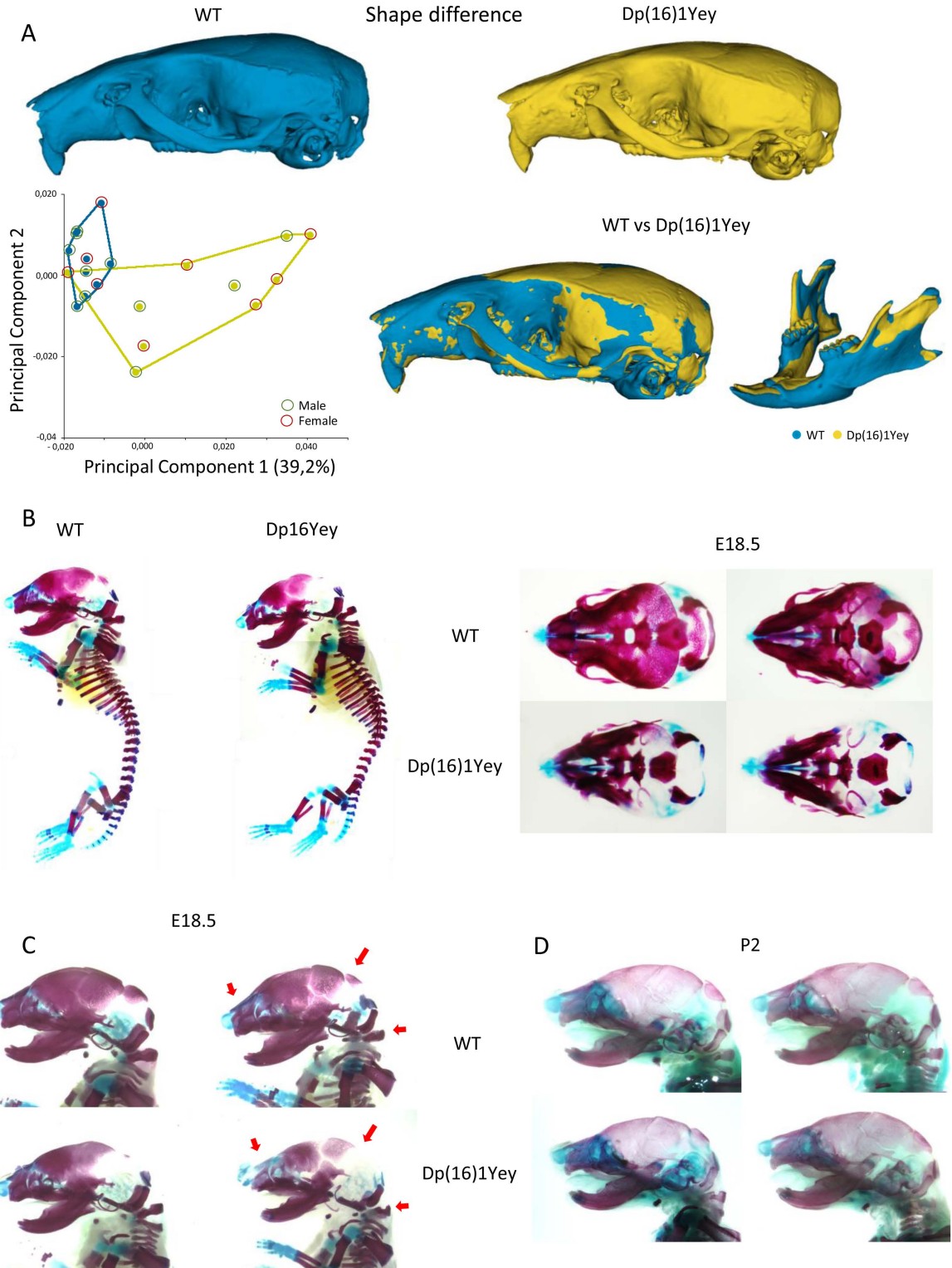

**Fig 1. Alteration of DS craniofacial features in Dp(16)1Yey is observed during late development.** (A) Morphometric analysis of the cranium and mandible of adult Dp(16)1Yey vs control. WT is in light blue and Dp(16)1Yey in yellow. PCA (first two components) of general Procrustes analysis of aligned cranium shapes, using data from females (red circle) and males (green circle) mice. The percentage of variance in PC1 corresponds to 39.2%.

In the figure WT vs Dp(16)1Yey shape difference warping, with in blue the bones with decreased dimensions in Dp(16)1Yey (midface hypoplasia and mandible), and in yellow, the bones with increased dimensions (neurocranium). (B-D) Skeletal staining with alizarin red and alcian blue. Comparison WT vs Dp(16)1yey E18.5 whole body, upper (top) and lower (bottom) magnification of the skull showing less mineralization in temporal, parietal, intraparietal, and occipital bones. (C) Magnification of lateral view of skeletal staining, WT vs Dp(16)1Yey at E18.5 (2 individuals). Red arrows showing less mineralization in nasal bones, neurocranium, and form defect in atlas vertebrae. (D) Lateral view of skeletal staining of two WT vs Dp(16)1Yey individuals showing normal ossification pattern at P2.

the Mmu16: Dp(16)7Yah, Dp(16)8Yah, Dp(16)14Yah, Dp(16)10Yah, Dp(16)11Yah, Dp(16)12Yah and Dp(16)13Yah. We analyzed their CF morphology and compared duplication versus their control wild-type (WT) littermates (Fig 2A).

Performing the identical craniometric analysis as previously described, we found significant skull changes in form and shape in all the models except Dp(16)8Yah (S2 Fig). Multivariable analysis using PCAs showed changes in the same direction along principal component 1 as seen in Dp(16)1Yey mice (Fig 2B). A significant contribution of the principal component 1 (PC1) for Dp(16)12Yah, Tg(Dyrk1a), was found as in Dp(16)1Yey, then Dp(16)7Yah and Dp(16)14Yah.

Similar changes in shape and form in the skull, with an overall reduction in midface region and a strong brachycephaly, were observed in Dp(16)1Yey, Dp(16)14Yah and Dp(16)12Yah (Fig 2B). The Dp(16)13Yah model also presented a reduction in the midface region. Still, the premaxilla and nasal region were not reduced, and the brachycephaly was not seen (Fig 2B). The Tg(Dyrk1a) model, overexpressing Dyrk1a alone, showed strong brachycephaly and reduced premaxilla and nasal bone.

In contrast, we found an inverse phenotype in Dp(16)7Yah compared to Dp(16)1Yey. We scored increased dimensions in the midface and decreased dimensions in the neurocranium; similar changes were found in Dp(16)11Yah but were less significant. Dp(16)10Yah presented an overall reduction of head dimensions, partially observed in the Dp(16)8Yah with a larger midface part. Still, an increase in premaxilla and occipital bone size, resulting in an elongation of the anteroposterior axis in Dp(16)10Yah also observed in Dp(16)13Yah (Fig 2B). The PCAs showed changes in the inversion direction along principal component 1 as seen in Dp(16)1Yey mice (Fig 2B).

For the mandibles, as in the skull, significant changes in form and shape were found in all the lines (S3 Fig). Here, the changes were different, although the most important changes were found in the same DS models, namely Dp(16)14Yah, Dp(16)13Yah, Dp(16)12Yah, and Dp(16)7Yah. In Dp(16)14Yah and Dp(16)12Yah, a reduction in the lateral width on the body of the mandible, ramus, molar and incisor alveoli, but an increase in the dimension in the condylar process (coincident with the brachycephaly found in the neurocranium) were detected; these changes were similar to the ones found in Dp(16)1Yey (S3 Fig). In the case of Dp(16)13Yah, this model also presented a similar phenotype but with increased dimensions in the ramus (apart from the condylar process) (S3 Fig). Dp(16)7Yah had increased dimensions in the body of the mandible, ramus, molar alveolus, and condylar process but maintained the reduced dimensions of the incisor alveolus. A similar phenotype is found in Dp(16)11Yah, but dimensions decrease in the angular process. Also, in Dp(16)10Yah, significant changes were found in the condylar process, presenting a reduction in the lateral width (Fig 2B).

The analysis of this new mouse panel allows us 1) to dissect the contribution of several Mmu16 region overdosages to CF phenotypes in DS and 2) to show the differences in their contribution to the cranium and mandible CF phenotypes.

**Candidate genes for the CF DS phenotype in DS rodent models.** We focused on the new chromosomal region selected from the previous morphometric analysis, encompassing 15 genes between *Setd4 –Dyrk1a* genes (Fig 3A). Inside this region, based on scientific literature and a data set from the FaceBase Consortium [65], we isolated a set of candidate genes. Among these genes, *Dyrk1a* (Dual Specificity Tyrosine Phosphorylation Regulated Kinase 1A) and *Ripply3* (*Ripply Transcriptional Repressor 3*) seemed promising targets for CF in DS. *Dyrk1a* overdosage in DS CF phenotype was already pointed out in several studies [50,32,66,67] while Ripply3 is a new interesting candidate redhead. Indeed, several studies demonstrated *Ripply3*'s role as a transcriptional repressor of *Tbx1* across species and developmental contexts [68,69,70]. In addition, as presented in the introduction, *Tbx1* is the primary genetic driver of CF

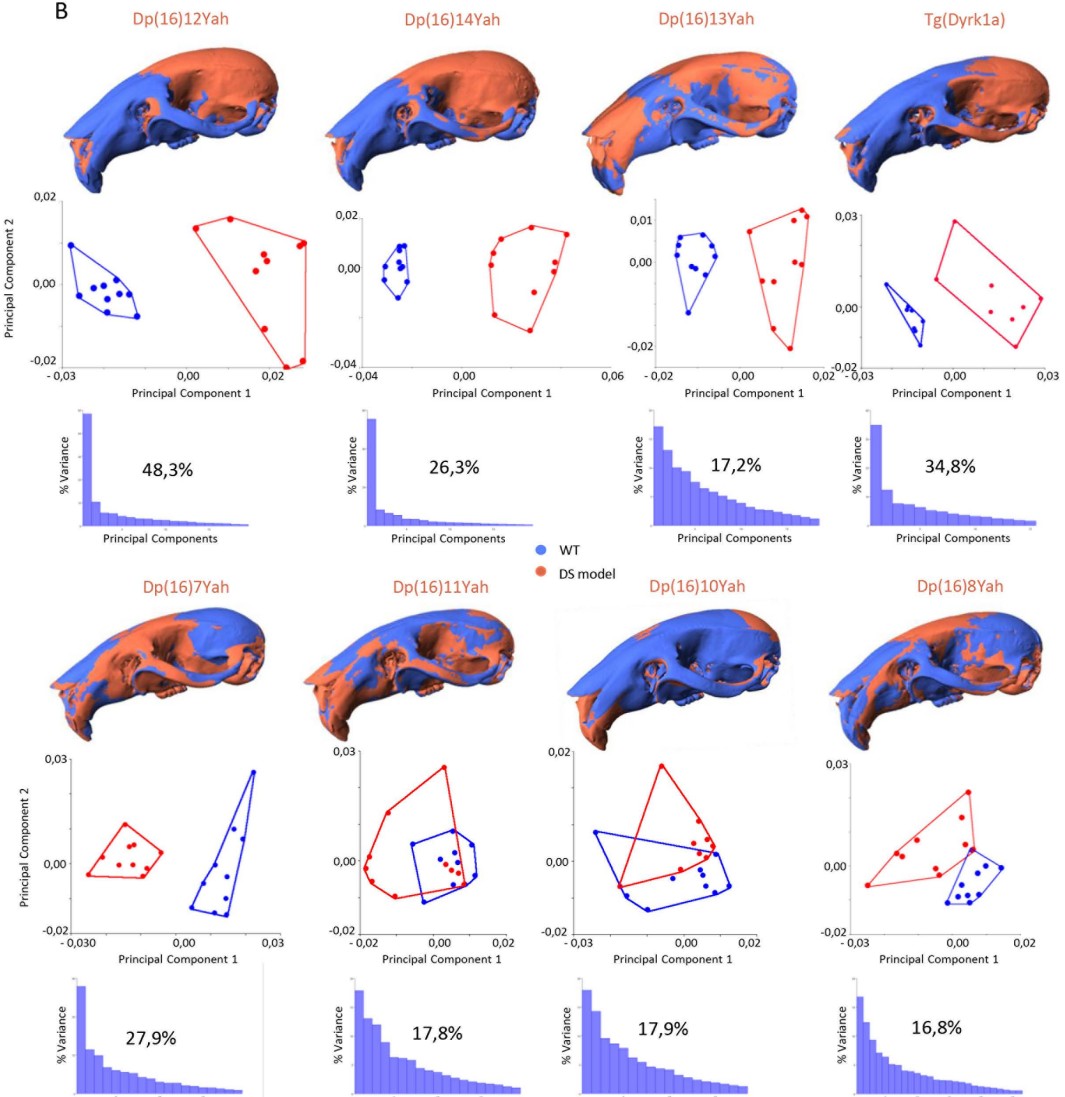

**Fig 2. New DS mouse model panel to identify genetic regions whose overdosage causes the craniofacial dysmorphology of Dp(16)1Yey.** (A) Summary table of a new panel of DS mouse model and their segmental duplication of the genetic interval. (B) Morphometric analysis of the cranium of the new panel of DS mouse models, plus Tg(Dyrk1a). Shape difference warping highlighted, in blue, the bones with decreased dimensions in DS models, and in red, the bones with increased dimensions. PCA (first two components) of general Procrustes analysis of aligned cranium shapes for every model with the percentage of variance graphic in PC1.

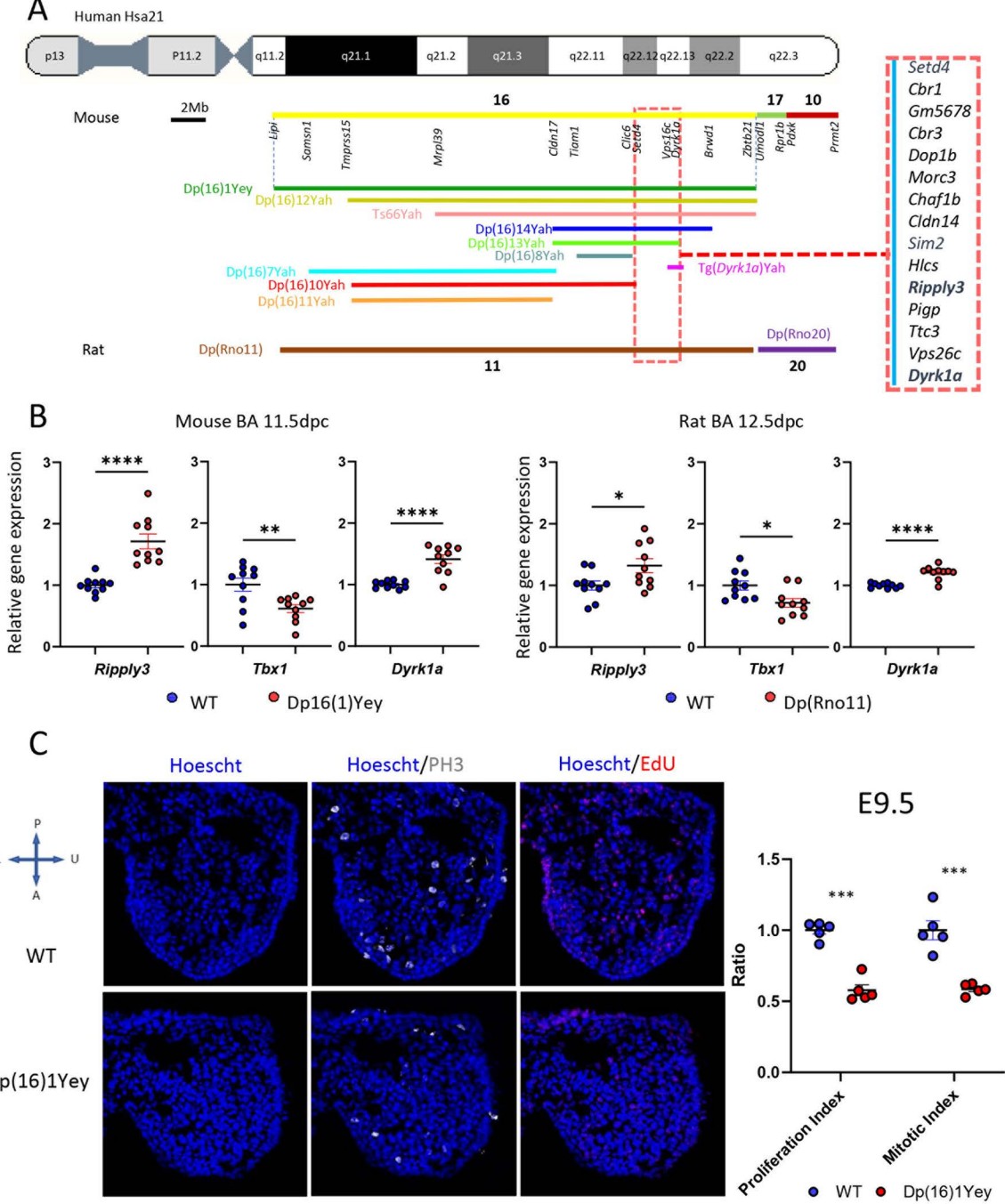

**Fig 3. Genetic mapping identifies a new chromosomic region and dosage-sensitive genes for the DS craniofacial phenotype.** (A) Scheme of the relative position of the new mouse and rat models in HSA21, and the new chromosomic region of interest for DS CF phenotype (red square). (B) RT-ddPCR results graphics for (left) Mouse model Dp(16)1Yey E11.5 in the left, and (right) rat DS Dp(RNO11) E12.5. RNA was isolated from the branchial arch region of trisomic and euploid controls (n = 10 per group). Relative gene expression ratios (trisomic/euploid) are shown for *Ripply3*, *Tbx1* and *Dyrk1a*.After confirming normality via Shapiro-Wilk test (α = 0.05), statistical comparisons were performed using a two-tailed unpaired t-test if the two groups had equal variance with the F-test; or a Welch's t-test if not. Significance thresholds: ****, p < 0.0001; ***, p < 0.001; ** for p < 0.01; *, p < 0.05. Significant overexpression of triplicated genes *Dyrk1a* and *Ripply3*, alongside downregulation of *Tbx1*, was observed in both species (vs. euploid controls). (C) Edu, PH3, and Hoechst unrevealed defects in the proliferation and mitosis of the NCC derivates in the 1st branchial arch during craniofacial development in Dp(16)1Yey at E.5. Here too, normality was confirmed by Shapiro-Wilk test (α = 0.05) and statistical comparisons were performed using a two-tailed unpaired t-test. In the right graphic, the proliferation (***, p < 0.000) and mitotic (***, p < 0.000) indexes were significantly reduced in Dp(16)1Yey.

phenotypes in DiGeorge Syndrome. Thus, we hypothesize that *Ripply3* overdosage in DS contributes to CF phenotypes in mouse models, through down-regulation of *Tbx1*

To explore this hypothesis, we analyzed the expression levels of *Dyrk1a, Ripply3*, and *Tbx1* via Droplet digital polymerase chain reaction (RT-ddPCR) in Dp(16)1Yey branchial arches and frontal process at E11.5. The stage is just after the neural crest-derived mesenchymal cells are differentiated into different bones in the facial region [71]. In the Dp(16)1Yey samples, an overexpression of *Dyrk1a* and *Ripply3* is detected in the craniofacial branchial arches in mutant versus wild-type. Concomitantly, *Tbx1* was downregulated in trisomic model versus control.

We also did similar expression studies in the Dp(RNO11) rat model with complete duplication of *Lipi-Zbtb21* region in the rat chromosome 11 [51] at embryonic stage E12.5 (homologous to the mouse stage). Similar results were found in the rat DS models with overexpression of *Dyrk1a* and *Ripply3* and down-regulation of *Tbx1* (Fig 4B). Overall, the *Ripply3* overexpression due to the triplication of the gene has the same consequence with a reduced expression of *Tbx1* in the branchial arches.

### Role of *Dyrk1a* overdosage in the increased dimensions in neurocranium (brachycephaly) on DS mouse models

*Dyrk1a* has been implicated in several DS phenotypes and craniofacial abnormalities [50,66,32,67] and in the last decade has become one of the top candidate gene in DS for therapeutic intervention [72,39]. Here we took advantage of Tg(*Dyrk1a*) - a model with 3 copies of *Dyrk1a* [50] - and compared with the results of the new model Dp(16)13Yah where Dyrk1a is not overexpressed, to confirm the role of *Dyrk1a* in the development of the DS CF phenotype. Brachycephaly and higher dimensions in the neurocranium were present in Tg(*Dyrk1a*), but also a reduction in the midface region, a phenotype very similar to the one observed in Dp(16)1Yey (Fig 2). In Dp(16)13Yah, a reduction in the midface region was also seen but focussed in the maxillary bones, the premaxilla and nasal region were not reduced and the strong brachycephaly was absent. Overall, this comparison confirmed the hypothesis and the role of *Dyrk1a* overdosage in inducing the brachycephaly.

### *Dyrk1a* and *Ripply3* overdosages affect the proliferation and mitosis of the NCC derivates in the first branchial arch during craniofacial development

In Tg(Dyrk1a) and in the new panel of mouse models (specifically in Dp(16)14Yah, Dp(16)12Yah and Dp(16)13Yah), significant changes were observed in the midface region, in structures that share the same embryonic origin, the neural crest cells (NCC) [73]. These findings, considering the relation of *Dyrk1a* with an altered expression of critical craniofacial regulators fundamental for cranial neural crest development [40] allows us to postulate that *Dyrk1a* and probably *Ripply3* overdosages could affect the proliferation of the NCC derivates during craniofacial development.

To demonstrate this, we monitored the proliferation of NCC with 5-Ethynyl-2'- deoxyuridine (EdU) a thymidine analogue incorporated into the DNA during replication [74]. Compared to controls, immunohistological analysis was used to detect the proliferation of NCC derivates in the first branchial arch of the Dp(16)1Yey embryos. In addition, quantitative analyses defined the proliferation and mitotic index [75]. A reduced proliferation and mitotic index in the first branchial arch were detected (Fig 3C). Thus, overexpression of triplicated genes from Dp(16)1Yey, including *Dyrk1a* and *Ripply3*, lead to the reduced proliferation of mesenchyme cells from the branchial arches.

### Confirmed role of *Ripply3* overdosage during midface development in DS mouse models

To confirm the role of *Ripply3* in midface hypoplasia, we crossed mice with a loss-of-function allele (*Ripply3*$^{tm1b/+}$) obtained from the IMPC initiative (www.mousephenotype.org; S4 Fig) with the Dp(16)1Yey line. We obtained Dp(16)1Yey/*Ripply3*$^{tm1b}$ males carrying a trisomy of all the genes present in Mmu16 but with only two functional copies of *Ripply3*, to rescue the midface hypoplasia of Dp(16)1Yey.

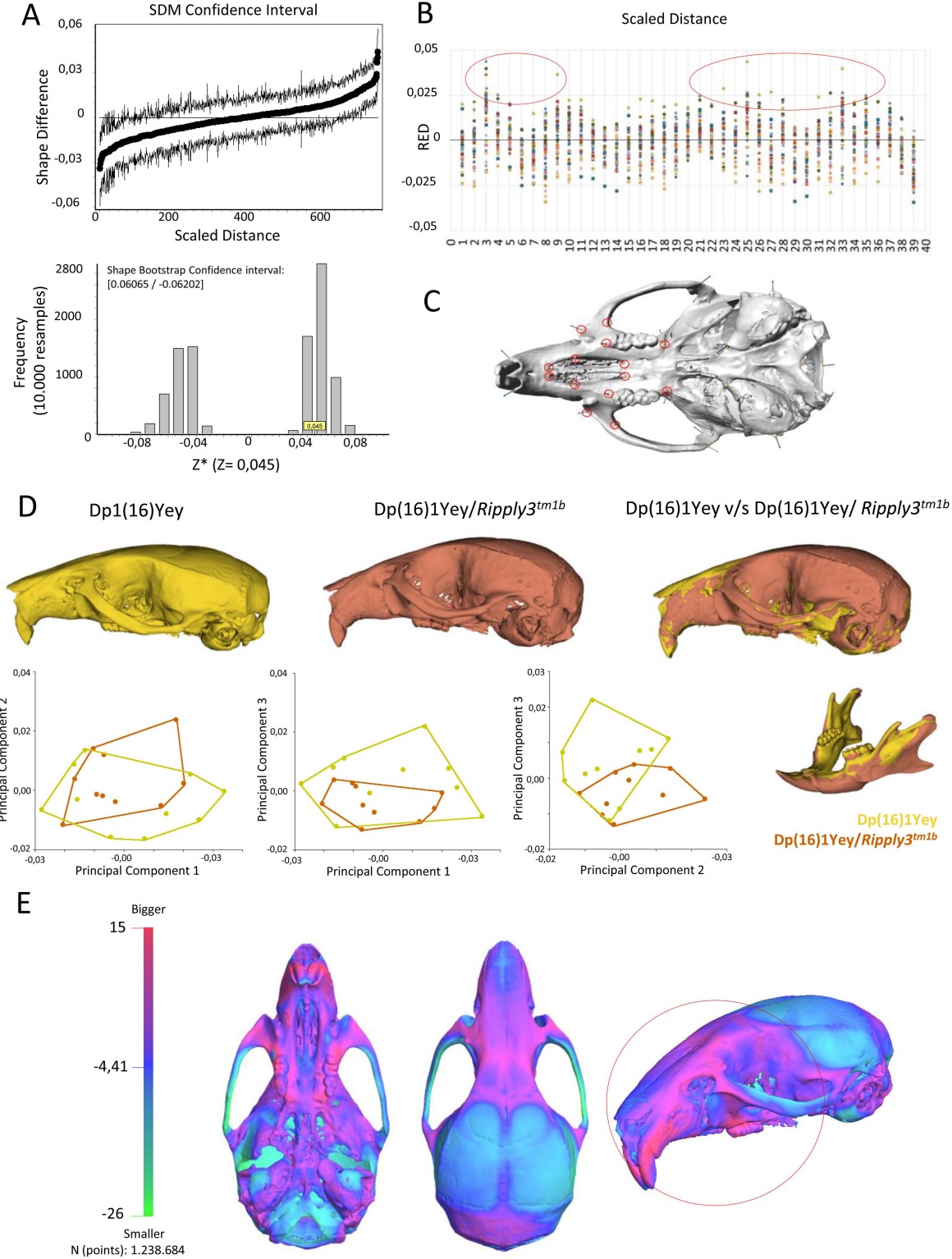

**Fig 4. Shape rescue in the midface phenotype in the Dp(16)1Yey/ _Ripply3_^tm1b compound mutant.** (A) Shape difference matrix based on Dp(16)1Yey vs Dp(16)1Yey/ _Ripply3_^tm1b comparison. Confidence interval and Frequency Bootstrap graphics with 10.000 resamplings showed significant shape changes (Z statistics = 0,045, CI = [0.06065/ -0.06202]). Influence landmarks analysis to show the most influential landmarks (red circle) that lead

to significant changes in Dp(16)1Yey/ Ripply3^tm1b vs Dp(16)1Yey in (B) with their position (C) located in the midface (3D model of Dp(16)1Yey/ *Ripply3^tm1b* mouse average model with landmarks, done with Stratovan Checkpoint). (D) Shape difference warping of Dp(16)1Yey vs Dp(16)1Yey/ *Ripply3^tm1b*. Dp(16)1yey is displayed in yellow and Dp(16)1Yey/ *Ripply3^tm1b* in orange. The bones with increased dimensions are in orange in the cranium figure of Dp(16)1Yey vs Dp(16)1Yey/*Ripply3^tm1b* warping. Maxillary bone, premaxilla, alveolar process and neurocranium are similar to that found in Dp(16)1Yey). In yellow, the bones with decreased dimensions (nasal bone and skull base). In the mandible of Dp(16)1Yey vs Dp(16)1Yey/ *Ripply3^tm1b* warping, the mandible presented the same changes found in Dp(16)1yey. PCA (first three components) of general Procrustes analysis of aligned cranium shapes. Showing significant changes in PC1/PC2, PC3/PC1 and PC3/PC2. (E) Comparison of Dp(16)1Yey average model vs Dp(16)1Yey/Ripply3^tm1b average model with a 3D heatmap from 3dMD Vultus software analysis. Pink/red shows increased shape dimensions in all the structures corresponding to the midface. In light blue, the structures with no significant changes. On the left, the histogram of every point distance evaluated (in this case, more than 1.238.684 points) and the surface differences with the color code for the increase-decrease dimensions (Red to green).

Morphometric analysis of Dp(16)1Yey/*Ripply3^tm1b* showed significant shape changes in the skull and mandibles compared to Dp(16)1Yey. In the SDM influence landmarks analysis, we can observe that the most influential landmarks leading to increased dimensions are located in the midface (Fig 4C). The skull voxel analysis showed the expected result with a similar change in the neurocranium to that present in Dp(16)1Yey. We noticed a phenotypical shape rescue in the midface, with increased dimension in maxillary bones, alveolar process, and premaxilla. The mandible presented the same changes found in Dp(16)1Yey (Fig 4D). To confirm this result and obtain further details, we mapped the surface differences in 3dMD Vultus software. Using the average model of each population, Dp(16)1Yey average model vs Dp(16)1Yey/*Ripply3^tm1b* average model, we performed a superimposition of the 3D models employing landmarks to have an exact matching (Fig 4E). Once the comparison was made, we obtained a histogram of every distance evaluated (in this case more than 1.238.684 points) and a 3D heatmap with the surface differences. In the heat map, in red, we found the structures with increased dimensions in Dp(16)1Yey/*Ripply3^tm1b*, that correspond to the bones located in the midface. In light blue, the structures that did not present significant changes (Fig 4E).

We also performed the morphometric analysis of Dp(16)1Yey/*Ripply3^tm1b* vs WT. As observed in S5A Fig, significant SHAPE changes were observed in the skull (S5A Fig) and the most influential landmarks leading to this significant shape changes were located in the neurocranium (S5B Fig). The Dp(16)1Yey/*Ripply3^tm1b* skull voxel-based analysis (S5C Fig) showed increased dimensions in the neurocranium, similar to the changes found in Dp(16)1Yey, but not significant changes in the midface structure, overlaid now by the WT structure. These data allow us to determine a phenotypical shape rescue in the midface in Dp(16)1Yey/*Ripply3^tm1b* due to rescue of *Ripply3* dosage in the Dp(16)1Yey.

## Integrative multivariate analysis of the craniofacial studies unravels four different subgroups of DS models

So far, after doing separate analyses of the DS models with their mutant and control littermates, we decided to combine all the data, performed a PCA analysis of the Euclidian distance and see whether we could discriminate different contributions (Fig 5 with PC1 and PC2 dimensions, and S6 Fig to show PC2 and PC3 dimensions). As shown in Fig 5, all the wild-type littermate controls, from the B6C3B F1 hybrid genetic background (for Ts66Yah) and the B6J pure background (for all the other models), clustered together for the cranium and the mandible, with a higher variation, probably due to the altered position of some landmarks.

Focusing on the skull, the DS models were divided into four main groups based on the first two dimensions. One is composed of the Dp(16)1Yey, Dp(16)1Yey/*Ripply3^tm1b*, a second of Dp(16)12Yah and Dp(16)14Yah, a third with Dp(16)10Yah, Dp(16)13Yah and Dp(16)8Yah, and the Dp(16)11Yah mixed with the wild-type while the Ts66Yah, the Tg(Dyrk1a) are on the same side of the DS models and the Dp(16)7Yah stayed apart from the other on PC1. Somehow, this distribution of models reflected the complexity of genetic interactions for the cranium phenotypes with at least 3 minimal Mmu16 regions: CF1 from *Setd4* to *Brwd1,* involving notably *Dyrk1a* and *Ripply3*, a second CF2 mapping to *Tiam1-Clic6*, and a third CF3 *Samsn1-Tmprss15* responsible for the Dp(16)7Yah phenotypes. Interestingly the Dp(16)1Yey, and the Dp(16)1Yey/*Ripply3^tm1b* cranium were well-separated in S6B Fig, certainly due to the *Ripply3* dosage rescue.

 

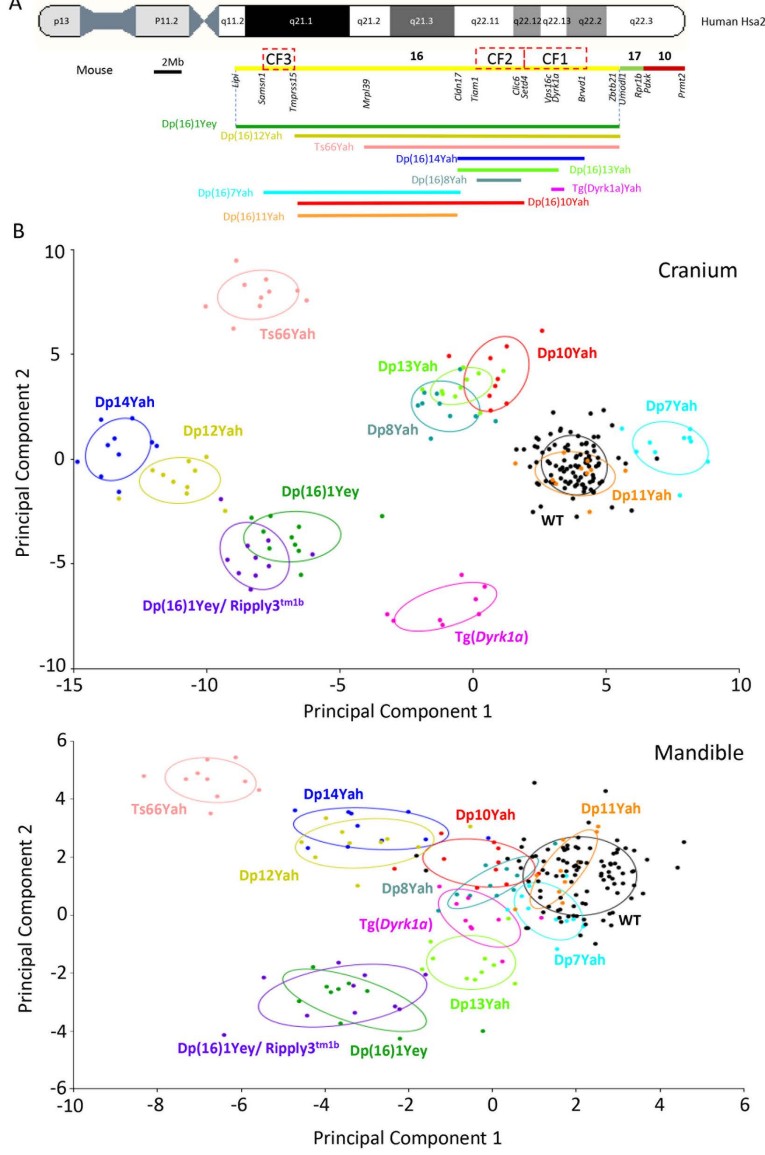

**Fig 5. Integrative multivariate analysis of the craniofacial studies.** (A) Schematic representation of DS models and their relative position to HSA21, showing the 3 new CF regions (red squares) involved in the DS CF phenotype located on Mmu16. (B) Integrative multivariate analysis of all the models used in this study, plus Ts66Yah vs wild-type. The PCAs correspond to a canonical variate analysis (Procrustes Distance Multiple Permutations tests at 1000 iterations). DS strains show significant differences compared with their wild-type controls. DS models were separated into four main groups with the cranium PCA graph, whereas for the mandible, the graph showed a prominent group of 5 models close to the wild-type and two branches separated on PC2.

For the mandible, the graph showed another complexity with a leading group of 5 models closed to the wild-type (Dp(16)11Yah, Dp(16)7Yah, Dp(16)10Yah, Dp(16)8Yah, Tg(Dyrk1a)) then two branches separated on PC2 with on one side the Dp(16)1Yey, Dp(16)1Yey/*Ripply3tm1b*, linked to the wt-like group with Dp(16)13Yah, and on the other side Dp(16)12Yah and Dp(16)14Yah. The Ts66Yah being kept alone. Interestingly the Dp(16)1Yey, Dp(16)1Yey/*Ripply3tm1b*, Dp(16)12Yah and Dp(16)14Yah, are closer for cranium changes, but they are well-separated for the mandibular

phenotypes (S6B Fig). These differences could come from various interacting regions contributing to the cranium and mandibular phenotypes.

## Discussion

In DS individuals, craniofacial dysmorphism is almost 100% penetrant, but the contributive genetic and developmental factors were unclear. The DS CF phenotype typically encompasses microcephaly, a small midface, a reduced mediolateral orbital region, reduced bizygomatic breadth, a small maxilla, brachycephaly (a relatively wide and broad neurocranium), and a small mandible [17].

We used Dp(16)1Yey to study this characteristic phenotype. This model carries a complete duplication of the Mmu16 region homologous to Hsa21 [28] and a previously well-described DS-like craniofacial phenotype [30]. Our results reproduce the same findings using a standard craniofacial analysis plus a new voxel analysis, where we could observe the principal changes in the skull and mandibles in 3D. We found that the changes were correlated with the human DS CF phenotype, making Dp(16)1Yey the appropriate model to study DS CF phenotype. Besides, we performed skeletal staining where we could identify a defect in intramembranous ossification, the results obtained at different embryonic stages allowed us to postulate that the changes observed at E18.5 did correspond to a delay and not to a continuous defect in intramembranous ossification. This defect could be related to a problem in the differentiation of mesenchymal cells into osteoblasts or in osteoblast proliferation with a subsequent attenuated osteoblast function [22].

Here we demonstrated that the craniofacial dysmorphism found in Dp(16)1Yey, correlated with the human DS CF phenotype, and we mapped three distinct chromosomic regions of Mmu16. Using a new panel of DS mouse models with specific CF phenotypes (Table 1), we could now identify new dosage-sensitive regions and genes responsible for DS CF phenotype. Saying this, the Ts66Yah behaved independently according to dosage effects in CF phenotypes compared to lines with segmental duplication. This may reflect a unique contribution of the minichromosome on the CF severity while the gene content is close to Dp(16)14Yah. Further experiments with new models will help to confirm the role of the segregating chromosome in CF DS phenotypes [76].

This report excludes the *Tmprss15-Grik1* regions, triplicated in Dp(16)11Yah alone, with almost no effect on CF form and shape. This agrees with the Dp(16)9Tyb lack of CF phenotype [32]. On one hand, two of the three CF regions defined here, CF1 and CF3, settled the telomeric regions described by Redhead et al. [32]. Nevertheless, we found the contribution of the most centromeric part CF3 involved in cranium enlargement as new. It could be slightly artificial as this effect was not seen in the Dp(16)1Yey, but could also be due to an effect specific to this

**Table 1. Summary of the CF phenotypes observed in DS models. Table expressing the level of changes found in skulls and mandibles: mild, strong and no significant (NS).**

|  | Skull - Cranium | | Lower jaw - Mandible | |
| --- | --- | --- | --- | --- |
|  | Shape | Form | Shape | Form |
| Dp(16)1Yey | Strong | Strong | Strong | Strong |
| Dp(16)1Yey/*Ripply3tm1b* | Mild | Mild | Mild | Strong |
| Tg(Dyrk1a) | Strong | Strong | Mild | NS |
| Dp(16)14Yah | Strong | Strong | Strong | Strong |
| Dp(16)12Yah | Strong | Strong | Strong | Strong |
| Dp(16)8Yah | Mild | NS | Mild | NS |
| Dp(16)10Yah | Mild | Strong | Mild | Strong |
| Dp(16)13Yah | Mild | Mild | Strong | NS |
| Dp(16)7Yah | Strong (Inverse) | Mild | Mild | Mild |
| Dp(16)11Yah | Mild (Inverse) | Mild | Mild | Mild |

region while not triplicated with CF2 and CF3. Only more detailed investigations with new models would allow us to discriminate the genetic interaction of the 3 CF regions. On the other hand, while the overdosage of *Dyrk1a* was crucial for the Dp(16)1Tyb phenotypes [32], the sole overexpression of *Dyrk1a* in the Tg(Dyrk1a) line only replicated well the skull phenotype, more precisely the brachycephaly. Conversely, using PCA, transgenic individuals were closer to the WT group for the lower jaw. Recently, *Dyrk1a* has been identified as one of the genes required in three copies to cause CF dysmorphology in mouse models of DS, and the use of DYRK1A inhibitors or genetic knock-out of DYRK1A has been shown to rescue the skull and jaw malformations [66,32]. However, [40], showed that a decrease in *Dyrk1a* in Xenopus resulted in craniofacial malformations, altered expression of critical craniofacial regulators as *Pax3* and *Sox9* fundamental for cranial neural crest development, and presented altered retinoic acid, hedgehog, nuclear factor of activated T cells (*NFAT*), *Notch* and *WNT* signaling pathways. These results indicate that DYRK1A function is critical for early craniofacial development and must properly regulate the expression of specific craniofacial regulators in the branchial arches [40]. We achieved to demonstrate, thanks to the analysis in Tg(*Dyrk1a*) and Dp(16)13Yah, that 3 copies of *Dyrk1a* are necessary to induce the brachycephaly found in DS. Thus, *Dyrk1a* overdosage is essential and sufficient for brachycephaly, but other genes are responsible for the mandibular phenotypes observed in DS [66].

Our other candidate gene, *Ripply3*, is a transcriptional corepressor, that acts as a negative regulator of the transcriptional activity of *Tbx1* and plays a role in the development of the pharyngeal apparatus and derivatives [70]. *Tbx1* is the first dosage-sensitive gene identified in models for the DiGeorge syndrome (DGS)/velocardiofacial syndrome (VCFS), a congenital disorder characterized by neural-crest-related developmental defects. In human and DGS models, *TBX1* haploinsufficiency causes craniofacial anomalies [41] and contributes to heart defects [77]. More precisely, the phenotype observed in the mutant mice for the T-box gene, *Tbx1*$^{+/-}$, encompasses abnormal development of the skeletal structures derived from the first and second pharyngeal arches, with reduced dimension of the midface [42]; a similar situation found in the DS mouse models. In Dp(16)1Yey at an early stage (E11.5), *Ripply3* is overexpressed, and consecutively, *Tbx1* is downregulated in the midface precursor tissues. Still, we also detected a defect in cell proliferation of the NCC derivates in the first branchial arch, which also demonstrated a contribution to the midface shortening. In addition, our new model Dp(16)1Yey/*Ripply3*$^{tm1b}$ demonstrated an increased shape dimension in the structures corresponding to the midface compared to Dp(16)1Yey and to wt control. Taken together, our hypothesis that *Ripply3* overdosage contributes to midfacial shortening in Down syndrome through the downregulation of *Tbx1* is supported by gene expression analyses in the developing branchial arches of both mouse and rat DS models, as well as by genetic dosage rescue experiments in the Dp(16)1Yey model.

Additionally, in Down syndrome (DS) mouse models, like Ts1Rhr and Ts1Cje, *Tbx1* gene expression showed significant downregulation in the brain, suggesting its potential role in delayed fetal brain development and postnatal psychiatric phenotypes associated with the condition [78]. Considering this information, we postulated that the overexpression of *Ripply3* in DS mouse models will lead to a downregulation of *Tbx1*, in other DS organs and tissues, leading to additional changes. As such, some DS heart defects, such as the tetralogy of Fallot observed in some individuals, may be related to *Ripply3*-dependent downregulation of *Tbx1*, whereas in DGS, they are caused by the direct *Tbx1* haploinsufficiency [77]. Interestingly, *Tbx1* down-regulation in DGS is known to affect the structure of the brain with an overall decrease in myelin in the fimbria, probably through reducing oligodendrocyte generation [79], in the amygdaloid complex and close cortical regions [80]. Concomitantly, *Tbx1* haploinsufficiency causes phenotypes in social interaction and communication, a tendency toward repetitive behaviour, impaired working memory, slower acquisition of spatial memory, and reduced cognitive flexibility [81,79,82]. To some extent, similar changes have been described in DS models and should now be investigated in more detail. Altogether, investigating treatment for DGS to reestablish a normal TBX1 function will also be of interest for Down syndrome, not only for the craniofacial but also for the heart and brain function.

## PLOS Genetics

## Supporting information

**S1 Table.  39 Cranium Landmarks for morphometric analysis.**
(DOCX)

**S2 Table.  22 Mandible Landmarks for Morphometric Analysis.**
(DOCX)

**S1 Fig.  Landmark detailed analysis of the cranium and mandibular phenotypes in Dp(16)1Yey.** (A) 3D model of wild-type mouse sample done with 3DSlicer software, showing the 61 Landmarks used in the CF analysis of Dp(16)1Yey (and in all the other DS models). 39 in the cranium and 22 in the mandible. (B) Form difference matrix analysis: FDM Bootstrap with 10,000 iterations showing significant changes in Form (p = 0.008). The FDM confidence interval graph shows a decrease of more than 90% of the distances measured. Form Influence landmarks graphic, showing the landmarks that present a relative Euclidean distance > 1.05 or < 0.95 (outside of the confidence interval 97,8%, red lines) and a general reduction of all dimensions.
(TIF)

**S2 Fig.  Form difference matrix analysis of the new panel of mouse models, plus Tg(Dyrk1a).** On the left, Cranium analysis using a difference matrix (FDM) was tested with Bootstrap (graph), showing significant changes in form for all models, except Dp(16)8Yah (p = 0.315), as indicated by the form difference interval graph. On the right, the same graph for the mandible shows significant changes in form for all the models except Dp(16)8Yah (p = 0.215). On the right, the FDM confidence interval graph shows the ratio of the distances measured.
(TIF)

**S3 Fig.  New DS mouse models mapping the location of dosage-sensitive genes that cause the craniofacial dysmorphology of Dp(16)1Yey (Mandibles).** Morphometric analysis of the mandibles of the new panel of DS mouse models, plus Tg(Dyrk1a) and Dp(16)1yey. Shape difference warping to display the mandible parts with decreased dimensions in DS models (in blue), and with increased dimensions (in red). PCA (first two components) of general Procrustes analysis of aligned mandible shapes for every model with the contribution to the explained variance for each dimension.
(TIF)

**S4 Fig.  Schematic representation of the *Ripply3<sup>tm1b</sup>* knock-out model derived from *Ripply3<sup>tm1a</sup>*.**
(TIF)

**S5 Fig.  Morphometric analysis of Dp(16)1Yey/Ripply3tm1b vs WT showed significant SHAPE changes at the level of the midface in the skull.** (A) The shape Distance matrix analysis showed statistically different distribution after bootstrap analysis. (B) the most influential landmarks leading to this significant shape changes are located in the neurocranium (red circles in the graphic and yellow circles in the scheme of the Skull). (C) The Dp(16)1Yey/Ripply3tm1b skull voxel-based analysis showed increased dimensions in the neurocranium, similar to the changes found in Dp(16)1Yey, but not significant changes in the midface structure, now mixing with the WT structure.
(TIF)

**S6 Fig.  PC2 and PC3 projection of the CF analysis discriminate the genotypes of DS models.** (A) Schematic representation of DS models and their relative position to HSA21. (B) PC2 and PC3 graphs for the cranium and mandible. Integrative multivariate analysis of all the models used in this study, plus Ts66Yah, vs wild-type. The PCAs (PC2 vs PC3) correspond to a canonical variate analysis (Procrustes Distance Multiple Permutations test at 1000 iterations). DS strains show significant differences compared with their wild-type controls. DS models were separated into four main groups in

the cranium graph, similar to PC1, but with distinct distributions. For the mandible, the graph showed a primary group of 5 models close to the wild-type and two branches separated on PC2, as observed in PC1 vs PC2.
(TIF)

## Acknowledgments

We thank the Mouse Clinical Institute (PHENOMIN-ICS) for helping maintain the mutant mouse models. Special thanks to Loïc Lindner and Pauline Cayrou for helping with ddPCR design and training, to Sophie Brignon, Charley Pinault, and Aurélie Eisenmann at PHENOMIN-ICS and the IGBMC animal facility for their services, and to Patrick Reilly for proof-reading the manuscript.

## Author contributions

**Conceptualization:** José Tomás Ahumada Saavedra, Agnes Bloch Zupan, Yann Herault.

**Data curation:** José Tomás Ahumada Saavedra, Agnes Bloch Zupan, Yann Herault.

**Formal analysis:** José Tomás Ahumada Saavedra, Claire Chevalier, Yann Herault.

**Funding acquisition:** José Tomás Ahumada Saavedra, Agnes Bloch Zupan, Yann Herault.

**Investigation:** José Tomás Ahumada Saavedra, Claire Chevalier, Agnes Bloch Zupan, Yann Herault.

**Methodology:** José Tomás Ahumada Saavedra, Claire Chevalier, Yann Herault.

**Project administration:** Agnes Bloch Zupan, Yann Herault.

**Resources:** Yann Herault.

**Software:** Yann Herault.

**Supervision:** José Tomás Ahumada Saavedra, Agnes Bloch Zupan, Yann Herault.

**Validation:** José Tomás Ahumada Saavedra, Agnes Bloch Zupan, Yann Herault.

**Visualization:** José Tomás Ahumada Saavedra, Agnes Bloch Zupan, Yann Herault.

**Writing – original draft:** José Tomás Ahumada Saavedra, Agnes Bloch Zupan, Yann Herault.

**Writing – review & editing:** José Tomás Ahumada Saavedra, Claire Chevalier, Agnes Bloch Zupan, Yann Herault.

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
