## [Decision Letter · Decision Letter 0]

4 May 2025

PGENETICS-D-25-00125

Ripply3 overdosage induces mid-face shortening through Tbx1 downregulation in Down syndrome models.

PLOS Genetics

Dear Dr. Herault,

Thank you for submitting your manuscript to PLOS Genetics. After careful consideration, we feel that it has merit but does not fully meet PLOS Genetics's publication criteria as it currently stands. Therefore, we invite you to submit a revised version of the manuscript that addresses the points raised during the review process.

Please submit your revised manuscript within 30 days Jun 03 2025 11:59PM. If you will need more time than this to complete your revisions, please reply to this message or contact the journal office at plosgenetics@plos.org. Please include the following items when submitting your revised manuscript:

We look forward to receiving your revised manuscript.

Kind regards,

Giovanni Bosco, Ph.D.

Section Editor

PLOS Genetics

Aimée Dudley

Editor-in-Chief

PLOS Genetics

Anne Goriely

Editor-in-Chief

PLOS Genetics

**Journal Requirements:**

At this stage, the following Authors/Authors require contributions: José Tomas Ahumada Saavedra, Claire Chevalier, Agnes Bloch Zupan, and Yann Herault. Please ensure that the full contributions of each author are acknowledged in the "Add/Edit/Remove Authors" section of our submission form.

The list of CRediT author contributions may be found here: https://journals.plos.org/plosgenetics/s/authorship#loc-author-contributions

https://journals.plos.org/plosgenetics/s/submission-guidelines#loc-parts-of-a-submission

4) We do not publish any copyright or trademark symbols that usually accompany proprietary names, eg ©,  ®, or TM  (e.g. next to drug or reagent names). Therefore please remove all instances of trademark/copyright symbols throughout the text, including:

- ® on pages: 7, 9, 18, and 32

- TM on pages: 10, and 12.

5) Thank you for including an Ethics Statement for your study. Please include:

i) The full name(s) of the Institutional Review Board(s) or Ethics Committee(s).

6) Please upload all main figures as separate Figure files in .tif or .eps format. For more information about how to convert and format your figure files please see our guidelines: 

7) We notice that your supplementary Figures, and Tables are included in the manuscript file. Please remove them and upload them with the file type 'Supporting Information'. Please ensure that each Supporting Information file has a legend listed in the manuscript after the references list.

8) Some material included in your submission may be copyrighted. According to PLOSu2019s copyright policy, authors who use figures or other material (e.g., graphics, clipart, maps) from another author or copyright holder must demonstrate or obtain permission to publish this material under the Creative Commons Attribution 4.0 International (CC BY 4.0) License used by PLOS journals. Please closely review the details of PLOSu2019s copyright requirements here: PLOS Licenses and Copyright. If you need to request permissions from a copyright holder, you may use PLOS's Copyright Content Permission form.

Potential Copyright Issues:

i) Figures 3C, 4C, and S1. Please confirm whether you drew the images / clip-art within the figure panels by hand. If you did not draw the images, please provide (a) a link to the source of the images or icons and their license / terms of use; or (b) written permission from the copyright holder to publish the images or icons under our CC BY 4.0 license. Alternatively, you may replace the images with open source alternatives. See these open source resources you may use to replace images / clip-art:

9) We note that your Data Availability Statement is currently as follows: "All the data are available as indicated in the manuscript.". Please confirm at this time whether or not your submission contains all raw data required to replicate the results of your study. Authors must share the “minimal data set” for their submission. PLOS defines the minimal data set to consist of the data required to replicate all study findings reported in the article, as well as related metadata and methods (https://journals.plos.org/plosone/s/data-availability#loc-minimal-data-set-definition).

10) Please amend your detailed Financial Disclosure statement. This is published with the article. It must therefore be completed in full sentences and contain the exact wording you wish to be published.

11) Please ensure that the funders and grant numbers match between the Financial Disclosure field and the Funding Information tab in your submission form. Note that the funders must be provided in the same order in both places as well.

12) Please provide a completed 'Competing Interests' statement, including any COIs declared by your co-authors. If you have no competing interests to declare, please state "The authors have declared that no competing interests exist". Otherwise please declare all competing interests beginning with the statement "I have read the journal's policy and the authors of this manuscript have the following competing interests:"

**Reviewers' comments:**

Reviewer's Responses to Questions

Reviewer #1: This study by Herault and colleagues capitalizes on mouse models of Down syndrome to identify driver genes for the craniofacial phenotype. Their approach to using chromosomal segments in the trisomy portion of human chromosome 21 is an elegant extension of the previous approach applied to human chromosome 22q11 (PMID: 16365290; PMID: 23917946; PMID: 19617637;PMID: 10545603; PMID: 16684884). The authors are also commended for rigorously controlling the genetic backgrounds of mouse models they used and generated, as this issue has compromised many mouse studies (PMID: 29369447).

Their analysis confirmed Dyrk1a and, additionally, discovered Ripply3 as driver genes for the phenotype. Remarkably, the authors also identified Tbx1 as a potential intermediate molecule, down regulation of which also has been shown to reduce the craniofacial phenotype. This is another clear evidence that even large CNVs contain single driver genes, refuting a claim made by some, based on a small-scale analysis, that each large CNV does not contain single driver genes.

As is often the case with work coming from this lab, the study is well designed, data are adequately analyzed and interesting, and the results are important and have broad implications. This reviewer has only minor suggestions for improving this excellent work.

1. The data presented in Figure 3BC show considerably variable variance among gene groups. It should be described in the figure legend if the homogeneity of variance was or was not violated among groups, if the former, what non-parametric tests were used for statistical analysis.

2. The images in Figure 3C are not of good quality, and EdU-positive nuclei are not visible. Higher-quality images should be used.

3. Data in Figure 3 are consistent with their hypothesis, but it does not directly demonstrate the causative role of Tbx1 as an intermediate variable between Dyrk1a/Ripply3 and craniofacial phenotypes. This limitation should be more fully discussed.

4. Tbx1 gene dose alterations contribute to the structural phenotypes of the brain (PMID: 34737458; PMID: 39463450) and affects the function of adult stem cells (PMID: 23917946). The authors might want to discuss this point.

Reviewer #2: This manuscript addresses the genetic causes of the craniofacial defects in Down syndrome (DS). The syndrome results from trisomy of chromosome 21 (Hsa21) and is thus most likely due to increased dosage of one or more of the genes on Hsa21. The authors use a mouse model of DS, Dp(16)1Yey, which has an extra copy of a region of mouse chromosome 16 (Mmu16) that is orthologous to Hsa21. They confirm earlier studies, showing that adult Dp(16)1Yey mice have altered craniofacial alterations including a smaller cranium and mandible, brachycephaly and midfacial hypoplasia. They then extend the analysis to a new series of 7 mouse strains with duplications of different parts of the Mmu16 region, in order to map the location of genes whose increased dosage causes the defects. They conclude that there are three areas of Mmu16 that contribute to the phenotype. In addition, they show that two of the causative genes are Dyrk1a and Ripply3. Of these, previous studies had already implicated increased dosage of Dyrk1a in causing craniofacial changes in DS. However, the identification of Ripply3 is a novel finding. Overall, this is an interesting study using suitable approaches to find the genes that cause DS phenotypes. However, I have several concerns, including that the manuscript is hard to follow in many places.

Major points.

1. The authors report that they find significant changes in Dp(16)1Yey mice in both form and shape. What do the authors consider to be the difference between form and shape?

2. How do the authors determine that there are statistically significant changes between Dp(16)1Yey mice and their WT controls?

3. Do the authors assess size changes separately from shape changes? I understand that they are measuring distances between landmarks shown in S1A. However, differences could be due to changes in size or shape or both. It would be important to distinguish these two distinct effects. Can size differences be regressed out to determine if there are shape differences as well?

4. Figure S1. What is the meaning of the graphs in B? They are not explained sufficiently well for the reader to understand. What is being plotted in each of these? What is the meaning of the x- and y-axes of the histogram? What is Form difference and Form Confidence Interval? Some of the labels are very small and hard to read. Same applies to Figures S2, 4A and B, which are also not understandable.

5. What data input is used for the PCA analysis shown in Figure 1A? Is it distances between landmarks? Or between landmarks and the centroid? Or something else? Please explain clearly. Are the differences between WT and Dp(16)1Yey significant? What statistical test is used for this? Provide suitable metrics, e.g. p-values.

6. The analysis of the multiple mouse strains in is hard to follow. The authors analyze 7 new mouse strains and present the data in Figures 2, S2, S3. A genomic map showing all the strains would be very useful at this point. In Figure 3A they authors show a map but it only contains 4 of these strains. A map with all the strains does not appear until Figure 5A, three figures later. And even this map is hard to read - the strains are not listed in either genomic order or in numerical order, with labels sometimes on the left and sometimes on the right, making it hard to find any specific strain.

7. On page 16 the authors announce that Ripply3 is a promising target for a gene that may be causing the craniofacial changes. However, they explain nothing about their logic for picking this gene - this is the first mention of the gene in the manuscript. Why was it chosen? Why were other genes ignored?

8. On page 18 the authors cross Dp(16)1Yey mice to a Ripply3 KO allele to test if reducing the copy number of Ripply3 reverses the defects. They state that Dp(16)1Yey/Ripply3tm1b mice show significant shape changes in the skulls and mandibles compared to Dp(16)1Yey mice. Where is the data showing that the shape differences are significant?

9. The PCA plots in Figure 4D show that the two genotypes overlap closely in shape. The legend says they are significantly different, but that seems surprising given the substantial overlap. What statistical test was used to show differences? Please include appropriate metrics, e.g. p-values.

10. Top of page 19 the authors state that the heat map shows increased dimensions in all the bones in the red region. Which genotype is increased relative to which other? Is it Dp(16)1Yey or Dp(16)1Yey/Ripply3tm1b that is increased? This confusion is present in several places in the manuscript which should be edited carefully.

Minor points.

11. Introduction. The authors state that Hsa21 has 671 genes, but then state that Ts65Dn mice are trisomic for 104 genes. I suspect that 671 refers to both coding and non-coding genes, whereas 104 refers to just coding genes. The authors should clarify this, and for consistency they might consider reporting numbers of the same type of genes for both human and mouse, e.g. just coding genes.

12. In the abstract and the discussion, the authors state that brachycephaly means a wider skull. Brachycephaly usually refers to the skull being shorter front to back.

13. Methods. Mouse strains are reported to have all been maintained on C57BL/6J. Was the Ts66Yah mouse also maintained on C57BL/6J? If not, could that be a reason why Ts66Yah skulls lie in a different part of the PCA plots from Dp(16)1Yey and others?

14. Results, page 13, paragraph 3. The final 2 sentences are repeated on page 14 paragraph 2.

15. Figures S4 and S5 are not cited in the manuscript.

16. Page 20, line 9. The authors state that Dp(16)1Yey mice have a complete duplication of Mmu16. This is not correct. They have a duplication of only 22.9 Mb of Mmu16.

**Have all data underlying the figures and results presented in the manuscript been provided?**

Reviewer #1: Yes

Reviewer #2: None

PLOS authors have the option to publish the peer review history of their article (what does this mean? ). If published, this will include your full peer review and any attached files.

**Do you want your identity to be public for this peer review?** For information about this choice, including consent withdrawal, please see our Privacy Policy .

Reviewer #1: No

Reviewer #2: No

**Figure resubmission:**
---

## [Decision Letter · Decision Letter 1]

17 Jul 2025

PGENETICS-D-25-00125R1

Ripply3 overdosage induces mid-face shortening through Tbx1 downregulation in Down syndrome models.

PLOS Genetics

Dear Dr. Herault,

Thank you for submitting your manuscript to PLOS Genetics. After careful consideration, we feel that it has merit but does not fully meet PLOS Genetics's publication criteria as it currently stands. Therefore, we invite you to submit a revised version of the manuscript that addresses the points raised during the review process.

Please submit your revised manuscript within 30 days Aug 16 2025 11:59PM. If you will need more time than this to complete your revisions, please reply to this message or contact the journal office at plosgenetics@plos.org. Please include the following items when submitting your revised manuscript:

We look forward to receiving your revised manuscript.

Kind regards,

Giovanni Bosco, Ph.D.

Section Editor

PLOS Genetics

Giovanni Bosco

Section Editor

PLOS Genetics

Aimée Dudley

Editor-in-Chief

PLOS Genetics

Anne Goriely

Editor-in-Chief

PLOS Genetics

**Journal Requirements:**

1) Please provide an Author Summary. This should appear in your manuscript between the Abstract (if applicable) and the Introduction, and should be 150-200 words long. The aim should be to make your findings accessible to a wide audience that includes both scientists and non-scientists. Sample summaries can be found on our website under Submission Guidelines:

https://journals.plos.org/plosgenetics/s/submission-guidelines#loc-parts-of-a-submission

2) We have noticed that you have uploaded Supporting Information files, but you have not included a list of legends. Please add a full list of legends for your Supporting Information files after the references list.

3) Please amend your detailed Financial Disclosure statement. This is published with the article. It must therefore be completed in full sentences and contain the exact wording you wish to be published.

4)  Please ensure that the funders and grant numbers match between the Financial Disclosure field and the Funding Information tab in your submission form. Note that the funders must be provided in the same order in both places as well.  

**Reviewers' comments:**

Reviewer's Responses to Questions

**Comments to the Authors:**

Reviewer #1: The authors are commended for thoroughly revising the manuscript.

One minor outstanding issue is that this reviewer asked about the assumption of homogeneity of variance of data presented in Figure 3, but the authors' reply discussed the assumption of normality. Please provide the results of Levene's tests to justify the use of parametric tests.

Reviewer #2: The authors have done a good job of responding to the various points I raised. The manuscript is easier to follow now.

I am left with two main points.

1. The authors cross the Dp(16)1Yey mice to mice with a deficiency in Ripply3. In the resulting mice (Dp(16)1Yey/Ripplytm1b) the dosage of Ripply3 is reduced from 3 to 2 and the authors use this strain to test if 3 copies of Ripply3 are required for the craniofacial phenotype. The authors then compare the skulls of Dp(16)1Yey v Dp(16)1Yey/Ripplytm1b mice and conclude that these are different (Fig 4 and S5), with a less reduced midface in Dp(16)1Yey/Ripplytm1b mice. Thus, they conclude that 3 copies of Ripply3 are required for the full phenotype. However, as the PCA plots in Fig 4 and S5 make clear, Dp(16)1Yey and Dp(16)1Yey/Ripplytm1b mice are still very similar, and apparently substantially different from WT controls. If correct, this implies that while Ripply3 contributes, it makes only a small contribution to the overall phenotype.

The authors should extend their analysis to a comparison of WT v Dp(16)1Yey/Ripplytm1b mice to determine if the latter are still substantially different from WT, or if the phenotype has been 'rescued'. This is an important point to make clear, because it may indicate that Ripply3 plays only a minor role in the craniofacial phenotype. If so, the conclusions in the title and abstract are probably overstated and should be toned down appropriately.

2. The authors state (title and abstract) that they have shown that Ripply3 exerts its effects on the craniofacial phenotype of Dp(16)1Yey mice by downregulating Tbx1. However, they provide no direct evidence for this, as they confirm in their response to Reviewer 1. At best this is a hypothesis supported by their Q-PCR data (reduced Tbx1 in Dp(16)1Yey mice). Thus both the title and abstract are misleading. The authors should tone down this conclusion and turn it into a possible hypothesis to be tested.

**Have all data underlying the figures and results presented in the manuscript been provided?**

Reviewer #1: Yes

Reviewer #2: Yes

PLOS authors have the option to publish the peer review history of their article (what does this mean? ). If published, this will include your full peer review and any attached files.

**Do you want your identity to be public for this peer review?** For information about this choice, including consent withdrawal, please see our Privacy Policy .

Reviewer #1: No

Reviewer #2: No

**Figure resubmission:**
---

## [Editor Report · Decision Letter 2]

5 Sep 2025

Dear Dr Herault,

We are pleased to inform you that your manuscript entitled "Ripply3 overdosage induces mid-face shortening through Tbx1 downregulation in Down syndrome models." has been editorially accepted for publication in PLOS Genetics. Congratulations! And sincere apologies for the delay in getting you this decision.

Yours sincerely,

Giovanni Bosco, Ph.D.

Section Editor

PLOS Genetics

Giovanni Bosco

Section Editor

PLOS Genetics

Aimée Dudley

Editor-in-Chief

PLOS Genetics

Anne Goriely

Editor-in-Chief

PLOS Genetics

Comments from the reviewers (if applicable):

**Data Deposition**

http://datadryad.org/submit?journalID=pgenetics&manu=PGENETICS-D-25-00125R2

**Press Queries**

---

## [Editor Report · Acceptance letter]

PGENETICS-D-25-00125R2

Ripply3 overdosage induces mid-face shortening through Tbx1 downregulation in Down syndrome models.

Dear Dr Herault,

We are pleased to inform you that your manuscript entitled "Ripply3 overdosage induces mid-face shortening through Tbx1 downregulation in Down syndrome models." has been formally accepted for publication in PLOS Genetics! Your manuscript is now with our production department and you will be notified of the publication date in due course.

With kind regards,

Anita Estes

PLOS Genetics

On behalf of:
